# Identification and validation of early genetic biomarkers for apple replant disease

**Annmarie-Deetja Rohr** [1]*, **Jessica Schimmel**[2], **Benye Liu**[3], **Ludger Beerhues**[3], **Georg Guggenberger**[2], **Traud Winkelmann**[1]

**1** Institute of Horticultural Production Systems, Woody Plant and Propagation Physiology Section, Gottfried Wilhelm Leibniz University Hannover, Hanover, Germany, **2** Institute of Soil Science, Gottfried Wilhelm Leibniz University Hannover, Hanover, Germany, **3** Institute of Pharmaceutical Biology, Technische Universität Braunschweig, Braunschweig, Germany

* rohr@baum.uni-hannover.de

**Data Availability Statement:** All data underlying this study are available under DOI 10.20387/bonares-kvak-frbf. The RT-qPCR data are accessible under DOI 10.20387/bonares-hrv8-3xf3.

## Abstract

Apple replant disease (ARD) is a serious threat to producers of apple trees and fruits worldwide. The ARD etiology is not unraveled and managing options are either economically not applicable or environmentally harmful. Thus, interest is given in biomarkers that allow to indicate ARD situations at early time points in order to classify soils according to ARD severity but also to analyze the effectiveness to potential countermeasures. This study aimed at (i) identifying ARD biomarkers on the transcriptional level in root tissue by analyzing the expression of previously identified candidate genes in ARD soils of different origin and texture and (ii) testing the specificity of these marker genes to ARD. *In vitro* propagated M26 plantlets were submitted to a bio-test with three ARD soils, either untreated or disinfected by γ-irradiation. Expression of seven candidate genes identified in a previous transcriptomic study was investigated by RT-qPCR in a time course experiment. Already three days after planting, a prominent upregulation of the phytoalexin biosynthesis genes *biphenyl synthase 3 (BIS3)* and *biphenyl 4-hydroxylase (B4Hb)* was observed in the untreated ARD variants of all three soils. The phytoalexin composition in roots was comparable for all three soils and the total phytoalexin content correlated with the expression of *BIS3* and *B4Hb*. The third promising candidate gene that was upregulated under ARD conditions was the *ethylene-responsive transcription factor 1B-like (ERF1B)*. In a second experiment M26 plantlets were exposed to different abiotic stressors, namely heat, salt and nutrient starvation, and candidate gene expression was determined in the roots. The expression levels of *BIS3* and *B4Hb* were highly and specifically upregulated in ARD soil, but not upon the abiotic stress conditions, whereas *ERF1B* also showed higher expression under heat stress. In conclusion, *BIS3* and *B4Hb* are recommended as early ARD biomarkers due to their high expression levels and their high specificity.

**Funding:** The work of ADR and TW was funded by the German Research Foundation (DFG, www.dfg. de) via the research training group GRK1798 "Signaling at the Plant-Soil Interface" and a grant to BL and LB (BE 1174/19-1). It was also part of the project BonaRes-ORDIAmur (JS, GG) funded by the German Federal Ministry of Research and Education (www.bmbf.de) within the frame of the program BonaRes (grant no. 031B0025B). The funders had no role in study design, data collection and analysis, decision to publish, or preparation of the manuscript.

**Competing interests:** The authors have declared that no competing interests exist.

## Introduction

Apple replant disease (ARD) presents a serious economic risk in orchards and tree nurseries where apple trees are frequently replanted [1, 2]. Characterized by its distinctive symptoms of reduced shoot growth, stunting, shortened internodes, root damage and root tip necrosis [3–5], ARD leads to reduced and delayed fruit yields as well as reduced fruit quality [1, 2]. The estimated yield losses caused by ARD range from 20–50% [6]. Eventually, ARD can render a site unprofitable for apple cultivation [7, 8]. Due to intensification of apple cultivation to certain areas accompanied by an increase of planting density, the problem of ARD has increased over the last decades. Crop rotation systems and soil exchange are usually difficult to employ in apple cultivation, mainly due to the high degree of technical specialization in apple growing sites and the perennial nature of the trees [9].

Numerous potential causal agents of ARD have been identified over the years. The growth-reducing effect of ARD can be abolished by disinfection of the affected soil, impressively demonstrating the biotic nature of the main causes of ARD [e.g. 10–12]. Among these an increase of detrimental oomycetes (*Pythium* [1, 2, 13] and *Phytophthora* [1, 13]) fungi (*Cylindrocarpon* [1, 2, 14–16] and *Rhizoctonia* [1, 13, 15]) and nematodes like *Pratylenchus penetrans* [10, 17–20] have been mostly attributed. Along with this, a decrease in beneficial soil organisms such as fluorescent *Pseudomonas* species has been reported [21, 22]. The abundance as well as the activity of these biotic factors can be influenced by abiotic factors such as soil texture, organic matter and pH [23, 24]. Overall, ARD presents a complex phenomenon which can best be characterized as a dysbiosis or negative plant-soil feedback. This is reflected in the most recent definition of Winkelmann et al. [9], which attributes the detrimental effect of the previous apple culture on the replant generation to a harmful shift in soil-borne (micro)organism communities.

The major counteractions employed against ARD are chemical soil disinfection (fumigation), biofumigation using *Brassicaceae* plant material, inter-row planting and soil substitution [9]. Because of its detrimental effects on the environment, chemical soil disinfection is not available any longer to several European countries due to its discontinued registration or its application is strictly regulated. Thus, an integrated solution to overcome ARD is of higher interest than ever before. Novel approaches aim at using *Brassicaceae* plant parts and seed meal for a biological relief from the ARD agents, anaerobic soil disinfestation or the incorporation of beneficial microorganisms to the soil to fight the disease [21, 25–30]. Apple genotypes with an increased tolerance towards ARD have been described [31, 32] and a long-term aim is breeding for ARD-resistant rootstocks. Until now, however, no feasible counteraction is available.

In order to develop sustainable countermeasures against ARD, its underlying complex causes have to be unraveled on the soil side but also inside the plant. Recently, Weiß et al. [11] examined the transcriptomic response in the roots of the ARD-sensitive apple rootstock M26 (hereafter indicated as M26) [10, 26] and observed a down-regulation of primary metabolism genes. At the same time, genes involved in secondary metabolite production, plant defense and regulatory and signaling genes were upregulated in response to ARD. Among the genes with a function in the secondary metabolism, those involved in phytoalexin biosynthesis were strongly upregulated. In addition, high amounts of the products of this biosynthesis pathway, namely 3-hydroxy-5-methoxybiphenyl, aucuparin, noraucuparin, 2-hydroxy-4-methoxydibenzufuran, 2'-hydroxyaucuparin and noreriobofuran, could also be detected in the ARD-affected roots [33]. These compounds are part of the biotic stress response and have been described to act in particular against fungal pathogens [34], which have been attributed an enhanced role in ARD [2, 16].

Interest is given in biomarkers that allow an early detection and possibly also a quantification of the reaction of apple to ARD. These biomarkers would not only support breeding approaches for ARD tolerance but also allow to evaluate the efficacy of newly developed management options. In this study, seven genes that were identified to be strongly regulated in apple roots upon contact with ARD-affected soil [11], i.e. *1-aminocyclopropane-1-carboxylate oxidase homolog 1-like*, *ethylene-responsive transcription factor RAP2-11-like*, *ethylene-responsive transcription factor 1B-like*, *gibberellin-regulated protein 1-like*, *zinc finger domain-containing protein 10-like*, *biphenyl synthase 3* and *biphenyl 4-hydroxylase*, were chosen to be tested for their suitability as potential transcriptional biomarkers for an early detection of ARD. Gene expression was analyzed in apple roots growing on three different either untreated or γ-irradiated ARD-affected soils to provide a comparison to disease-free conditions, and was complemented by phytoalexin analysis. In a first experiment, the comparison of ARD-affected soils of different soil texture and chemical characteristics allowed distinguishing universal ARD responses from soil-specific responses regarding candidate gene expression. The promising biomarker candidates resulting from this approach were then tested in a second experiment for their specificity. Therefore, apple plants were exposed to different abiotic stressors, such as heat, salt stress, and nutrient starvation, and candidate gene expression was determined in the roots. Since the disinfected γ-irradiated ARD-affected soil can be regarded as a rather artificial control, the specificity test also included virgin (healthy) soils from plots adjacent to three ARD plots were included as an additional control.

## Material and methods

### Soil origin and disinfection

For the first experiment (candidate gene identification), soils from the three BonaRes-ORDIAmur ARD reference sites (www.ordiamur.de) in Germany were sampled in a depth of 0–20 cm in January 2017: Ruthe (Leibniz University Hannover, Sarstedt, 52˚14'39.8"N 9˚49'08.2"E), Heidgraben (Baumschule Harald Klei, Heidgraben, 53˚41'57.5"N 9˚40'59.6"E), and Ellerhoop (Gartenbauzentrum Schleswig-Holstein, Ellerhoop, 53˚42'51.9"N 9˚46'13.0"E). These soils were chosen to represent different soil textures. The Ruthe soil is of a loamy texture, while soil from Heidgraben is of a very sandy consistency and Ellerhoop soil is a loamy sand [12]. The soils were homogenized by sieving through an 8-mm mesh. One aliquot of each soil volume was packed in autoclavable plastic bags at a volume of 12 L each and sent for γ-irradiation with a minimum dose of 10 kGy (recorded dosages: 11.16 kGy minimum, 32.81 kGy maximum, Beta Gamma Service, Wiehl, Germany). The remaining untreated soil was transferred to buckets, covered with breathable MyPex fabric (Don & Low Limited, Angus, Scotland) and stored at outdoor temperature for approximately one month during which the γ-irradiation took place. The untreated soil will be referred to as ARD and the γ-disinfected soil as γARD. At the start of the experiment, all soil variants were supplemented with 2 g L$^{-1}$ of the slow-release fertilizer Osmocote Exact Standard 3–4 M (16% total nitrogen, 9% phosphorus pentoxide, 12% potassium oxide, 2% magnesium oxide + trace elements, Everris International B.V., Geldermalsen, The Netherlands, https://icl-sf.com/global-en/products/ornamental_horticulture/8840-osmocote-exact-standard-3-4m).

In the second experiment (candidate gene validation), salt and heat stressed plants were potted into a mix of peat substrate and sand (Steckmedium, Klasmann-Deilmann GmbH, Geeste, Germany, with perlite (1–1.7 mm), white peat (0–7 mm) and white sod peat (1–7 mm) + sand, 2 + 1, 60 mg L$^{-1}$ nitrogen, 70 mg L$^{-1}$ phosphorus, 120 mg L$^{-1}$ potassium, 85 mg L$^{-1}$ magnesium, 60 mg L$^{-1}$ sulfur). Peat substrate mixed with sand without any additional stress served as a control. All substrate variants (heat, salt stress and peat substrate) were fertilized

**Table 1. Overview of the treatments utilized for experiment 1 and 2 (candidate gene validation).** Fertilized variants were supplemented with 2 g L$^{-1}$ Osmocote Exact 3–4 M before filling of the pots. Peat substrate refers to Steckmedium by Klasmann-Deilmann GmbH, Geeste, Germany (see above).

| | Abbreviation | Substrate / soil | Treatment / specifications |
|---|---|---|---|
| Experiment 1 | ARD Ellerhoop | ARD soil Ellerhoop | Fertilized |
| | γARD Ellerhoop | ARD soil Ellerhoop | γ-irradiated, fertilized |
| | ARD Heidgraben | ARD soil Heidgraben | Fertilized |
| | γARD Heidgraben | ARD soil Heidgraben | γ-irradiated, fertilized |
| | ARD Ruthe | ARD soil Ruthe | Fertilized |
| | γARD Ruthe | ARD soil Ruthe | γ-irradiated, fertilized |
| Experiment 2 | ARD | ARD soil Heidgraben | Fertilized |
| | Grass | Control soil Heidgraben | Fertilized |
| | Peat (substrate) | Peat substrate + quartz sand (2 + 1) | Fertilized |
| | Heat | Peat substrate + quartz sand (2 + 1) | fertilized, plants 3 days at 37˚C |
| | Salt | Peat substrate + quartz sand (2 + 1) | one-time application of 50 mL 0.17 M NaCl (10 g L$^{-1}$) at the start of the experiment, fertilized |
| | Nutrient starvation | Quartz sand | not fertilized |

with 2 g L$^{-1}$ Osmocote Exact Standard 3–4 M as described above. Nutrient starvation was applied by potting the plants into unfertilized quartz sand. Further variants included soils from ARD and grassland plots from Heidgraben as references representing field conditions. In the grassland plots within the reference sites, no members of the Rosaceae had been grown. Thus, this soil served as a control soil that was not affected by replant disease but contained microorganism communities native to the site. The soils were sieved and fertilized as described above. All variants of experiments 1 and 2 with their respective substrates and stress treatments are depicted in Table 1.

The sampling and evaluation schedules of for both experiments are presented in S1 Table.

## Plant cultivation and sampling

For both experiments, plants of the ARD sensitive apple rootstock M26 were clonally propagated *in vitro* via axillary shoots on a modified MS medium [35] containing 3% sucrose, 0.5 μM indole-3-butyric acid (IBA) and 4.4 μM 6-benzylaminopurine (BAP). They were grown at 24˚C and 16 h light / 8 h darkness provided by Philips MASTER TL-D 58W/865 fluorescence tubes at a photosynthetically active photon flux density (PAR) of 25–30 μmol m$^{-2}$ s$^{-1}$. *In vitro* rooting was induced on ½ MS medium supplemented with 2% sucrose and 4.92 μM (IBA) [11]. The rooted plants were transferred to substrate for cutting propagation (Steckmedium, Klasmann-Deilmann GmbH, see above) and kept under a foil tent in the greenhouse for acclimatization. After two weeks, subsequently increasing ventilation of the tent was started until the plants were fully adapted to greenhouse conditions.

After approximately four weeks in the greenhouse the plants were transferred into the test soils (see Table 1). The first experiment used the ARD soils from the three sites Ruthe, Heidgraben and Ellerhoop (see above) either untreated or disinfected by γ-irradiation. The second experiment aiming at specificity testing used either ARD soil from Heidgraben, peat substrate (Steckmedium, Klasmann-Deilmann GmbH + sand, 2 + 1) or sand alone (Table 1). The different variants were prepared as follows: For salt stress, the plants were potted into fertilized peat substrate (Steckmedium, Klasmann-Deilmann GmbH + sand, 2 + 1, + 2 g L$^{-1}$ Osmocote) and initial watering was carried out with 50 mL of a 0.17 M NaCl solution per pot. From then on, irrigation was carried out with regular tap water. For the heat treatment, the plants were potted into fertilized peat substrate and grown for four days in the greenhouse. Thereafter, they were transferred to a culture cabinet (Rubarth Apparate GmbH, Laatzen, Germany) for three days

at 37˚C and a 16 h photoperiod, after which they were placed back into the greenhouse. Nutrient starvation was induced to the plants with unfertilized quartz sand. Controls included soil from the Heidgraben reference site from the ARD patches (ARD, positive control) and from patches covered in grass (Grass soil, negative control) as well as fertilized peat substrate-sand mix (substrate control, second negative control).

Round pots of 0.46 L volume and 10.5 cm diameter were used for the plants sampled after 1, 3, and 7 days in the first experiment and all plants in the second experiment. Additionally, 1 L pots were used for plants sampled after 8 weeks in the first experiment. All pots were lined with MyPex fabric to avoid washing out of soil or substrate during irrigation. They were placed in the greenhouse in a randomized design. Cultivation during the first three days was carried out without additional lighting. From then on, additional light was provided by SON-T Philips Master Agro 400 W lamps (Hamburg, Germany) if solar radiation fell below 25 klx to provide 16 h of daylight and thus comparable growing conditions over the whole year. The temperature in the greenhouse chamber was 21.1 ± 1.3˚C and the relative air humidity 58.1 ± 7.9% during the first experiment and 22.6 ± 3.1˚C and 64.2 ± 10.6%, respectively, for the second experiment. Plants were irrigated by hand on a daily basis and plant protection was carried out according to horticultural practice.

For the first experiment (candidate gene identification), complete root systems were harvested for gene expression analysis at day 0 (acclimatized plants before potting) and 1, 3 and 7 days after potting into the experimental soils. For this, 20 plants per soil variant and sampling day were selected randomly. They were unified into four pooled samples consisting of five plants each (S2 Table). For the second experiment (candidate gene validation), sampling was carried out after 7 and 14 days. Complete root systems of 15 plants per variant and time point were harvested for gene expression analysis yielding five pooled samples consisting of three individual plants each (S3 Table). After the seven-day sampling point in the second experiment, the heat-stressed plants were moved to the greenhouse until the end of the experiment. Therefore, these plants were no longer exposed to the heat at the second sampling point.

The plants were quickly but gently washed with tap water, blotted dry briefly and the complete root system was separated from the shoot, transferred to 2 mL reagent tubes (Sarstedt, Nümbrecht, Germany) and frozen in liquid nitrogen. Storage took place at -80˚C until RNA extraction. In both experiments, the shoot length was measured weekly. At the final day of sampling, the roots were separated from the shoots and the fresh masses of shoots and roots were recorded. Shoots and roots were then frozen at -20˚C and freeze dried for three days (Christ ALPHA 1–4 LSC, Osterode, Germany) to determine the dry mass. The freeze-dried samples were stored above silica gel (Carl Roth, Karlsruhe, Germany) until preparing for phytoalexin analysis (see below). In the second experiment, dry mass was recorded after oven-drying for three days at 80˚C.

## RNA extraction and first strand cDNA synthesis

Within the variants, pooling for both experiments (experiment 1: n = 4 pooled samples = biological replicates, see S2 Table, experiment 2: n = 5 pooled samples = biological replicates, see S3 Table) took into account that the mean shoot length of the pooled plants was comparable. The pooled root systems were homogenized at 29 Hz for 1 min using a mixer mill (Mixer Mill MM400, Retsch, Haan, Germany) cooled with liquid nitrogen. RNA extraction from 100 mg fresh mass of ground root powder was carried out with the InviTrap Spin Plant RNA Mini Kit (Stratec, Birkenfeld, Germany) according to the manufacturer's instructions. The included extraction buffer for phenol-containing plants was used (RP lysis buffer) and 40 μL of the provided elution buffer. Genomic DNA was removed from the RNA via digestion

with DNase I (Thermo Scientific, Waltham, MA, USA) following the manufacturer's instructions. The concentration and quality of the obtained RNA was determined spectrophotometrically (NanoDrop 2000c, Peqlab, Erlangen, Germany) and the quality was checked on a 1% agarose gel. The isolated RNA was stored at -80˚C until first strand cDNA synthesis using the RevertAid First Strand cDNA Synthesis Kit (Thermo Scientific, Waltham, MA, USA) with an input of 1 µg RNA and with random hexamer primers. The resulting cDNA was aliquoted for the qPCR measurements and stored at -20˚C until then.

## Quantitative PCR

All reactions were carried out on a CFX Connect™ cycler (Bio-Rad, Hercules, CA, USA) using the SYBR Green Supermix (Bio-Rad, Hercules, CA, USA). The primers used (Table 2) had been previously validated for the apple rootstock genotype M26 by Weiß et al. [11]. As reference genes, elongation factor 1-α (*EF1a*), elongation factor 1-β (*EF1b*) and tubulin beta chain (*TUBB*) [11] were selected after testing their expression stability between ARD and γARD variants (Table 2) at each time point for experiment 1 and between all six variants in experiment 2. Each forward and reverse primer were used in a concentration of 20 nM and amplification efficiency was determined for each experiment separately. A pool of all cDNA samples in equal amounts was created and the quantification cycles ($c_q$s) for the dilutions 1:5, 1:10, 1:50, 1:100, 1:500, 1:1000 and 1:2000 were determined. The protocol used for the efficiency tests and the following qPCRs was as follows: 3 min at 95˚C, followed by 10 s at 95˚C and 30 s at 60˚C for 40 cycles and a melt curve analysis (65˚C to 95˚C for 5 s each with an increment of 5˚C). The amplification efficiencies for each primer pair were calculated with the CFX manager

**Table 2. Primer sequences and amplicon lengths of candidate and reference genes used in RT-qPCR analyses [11].** MDP ID: Malus domestica predicted gene ID [38]. Amplification efficiency in % (E [%]) with corresponding coefficient of correlation ($R^2$). n.a. = not analyzed, excluded from the second experiment.

| Gene name (MDP ID) | Abbreviation | Primer sequence 5'– 3' | Amplicon length [bp] | experiment 1 | | experiment 2 | |
|---|---|---|---|---|---|---|---|
| | | | | E [%] | $R^2$ | E [%] | $R^2$ |
| *1-aminocyclopropane-1-carboxylate oxidase homolog 1-like* (MDP0000314499) | *ACO1* | f: CGCAGTTGGAGATGAACTTG | 167 | 98.2 | 0.995 | n.a.[1] | n.a. |
| | | r: CATGCCGTGATGGACAGTAG | | | | | |
| *ethylene-responsive transcription factor RAP2-11-like* (MDP0000177547) | *ERF RAP2.11* | f: TTCCAACAGCCGAAGCAAG | 169 | 71.3 | 0.980 | n.a. | n.a. |
| | | r: CTTTGATCTCAGCAACCCATCTC | | | | | |
| *ethylene-responsive transcription factor 1B-like* (MDP0000127134) | *ERF1B* | f: GTCACCTGAATCTTCGTTTG | 121 | 90.8 | 0.994 | 96.5 | 0.986 |
| | | r: GGAAATCAGACCGTAGAGAAG | | | | | |
| *gibberellin-regulated protein 1-like* (MDP0000140078) | *GASA1* | f: CGTTGCAGCTGTGTTCCTC | 156 | 92.5 | 0.993 | n.a. | n.a. |
| | | r: CATCTGCATGCCCGAATATGAG | | | | | |
| *zinc finger domain-containing protein 10-like* (MDP0000922823) | *GATAD10* | f: GCTCGTTTCTGGAGGAGTC | 153 | 91.3 | 0.983 | n.a. | n.a. |
| | | r: GATTCCCGCTGTCGTAGAATC | | | | | |
| *biphenyl synthase 3* (MDP0000287919) | *BIS3* | f: GGCAAGAAGCAGCATTGAAAG | 105 | 97.8 | 0.997 | 94.6 | 0.998 |
| | | r: CACAACCTGGCATGTCAAC | | | | | |
| *biphenyl 4-hydroxylase* (MDP0000152900) | *B4Hb* | f: GCTGAGTATGGCCCGTATTG | 156 | 98.7 | 0.996 | 96.7 | 0.999 |
| | | r: AGGAACCCGTCGATTATTGG | | | | | |
| *elongation factor 1-alpha* (MDP0000304140) | *EF1a* | f: GAACGGAGATGCTGGTATGG | 159 | 94.5 | 0.997 | 94.9 | 0.998 |
| | | r: CCAGTTGGCTCCTTCTTCTC | | | | | |
| *elongation factor 1-beta 2-like* (MDP0000903484) | *EF1b* | f: GAGAGTGGGAAATCCTCTG | 138 | 100.1 | 0.994 | 95.4 | 0.998 |
| | | r: ACCAACAGCAACCAATTTC | | | | | |
| *tubulin beta chain* (MDP0000951799) | *TUBB* | f: TTCTCTGGGAGGAGGTACTG | 147 | 99.8 | 0.998 | 90.6 | 0.998 |
| | | r: GTCGCATTGTAAGGCTCAAC | | | | | |

software version 3.0 according to Pfaffl [36] for both experiments separately. Only primers with an efficiency between 90 and 110% [37] were used for the following analyses leading to an exclusion of ethylene-responsive transcription factor RAP2-11-like (*ERF RAP2.11*) (Table 2). Gene expression analysis was performed in two technical and four (experiment 1) or five (experiment 2) biological replicates per time point and soil variant. Normalized gene expression was calculated according to Pfaffl [36].

## Phytoalexin extraction and analysis

Complete root systems were harvested at different time points to measure total phytoalexin contents and correlate them to expression of genes involved in phytoalexin biosynthesis and growth parameters (S1 Table). The samples for phytoalexin analysis were taken as follows for experiment 1: At days 0, 3, 7, 10 and after 8 weeks (day 56) in replicates of 10 single plants per treatment, which were later unified to obtain a minimal dry mass of 42.2 mg for the analyses (S2 and S4 Tables). For the second experiment, subsets of the 14 days gene expression samples (n = 5 pooled samples) were analyzed (n = 5 individual plants, S3 Table). The dried roots were ground to a fine powder using a mixer mill (Mixer Mill MM400, Retsch, Haan, Germany) at 29 Hz for 1 min.

Extraction of total phytoalexins was carried out as described previously [39]. In short, 1 mL of methanol supplemented with 50 µg of 4-hydroxybiphenyl (internal standard for relative quantification) were added to each samples in a 2 mL reagent tube. The samples were vortexed continuously with Vortex Genie 2 (Scientific Industries, Bohemia, NY, USA) at the maximum speed of 2.700 rpm. The resulting extracts were centrifuged at 13,000 rpm for 10 min and the supernatant was transferred into a new tube. A 200 µL aliquot of supernatant was transferred to a new 1.5 mL reagent tube. After drying under an air stream, the residue was re-dissolved in 200 µL ethyl acetate and centrifuged at 13,000 rpm for 10 min. The clear supernatant was transferred to a GC-MS vial with glass inlet. The ethyl acetate was evaporated by air stream and the residue was silylated with 50 µL N-methyl-N-(trimethylsilyl)trifluoroacetamide (MSTFA; Alfa Aesar, Thermo Fischer, Kandel, Germany) at 60˚C for 30 min. Silylated samples were analyzed by gas chromatography—mass spectrometry (GC-MS) using the following temperature program: 70˚C for 3 min, then linear increase of temperature from 70˚C to 310˚C over 24 min (10˚C min$^{-1}$) and finally 310˚C for 5 min. Helium was the carrier gas with a flow rate of 1 mL min$^{-1}$. The injection volume was 1 µL with split ratio 1:10. One technical replicate was measured per sample (biological replicate) after testing the stability of the system with two injections per sample. Furthermore, technical reproducibility was ensured by repeated measurements of the alkane standards at the beginning and the end of the sequences and additionally in the middle if the sequence contained more than 40 samples. At the same time, the alkanes served to calculate the retention index (RI). Quantification of individual compounds was done based on the internal standard 4-phenylphenol. A response factor of 1 was assumed for all compounds.

## Nutrient analysis

Nutrients were analyzed in the oven-dried leaves of plants from the second experiment sampled after 8 weeks. Leaves from the top, middle and bottom region of each shoot were collected. Material from one plant each was collected, which yielded 5 biological replicates, except for salt stress, where only four plants were available (S3 Table). Nutrients in each of these samples were measured in one technical replication. The samples were ground to a fine powder using 50 mL grinding vessels of a mixer mill (Mixer Mill MM400, Retsch, Haan, Germany) at 30 Hz for 1 min. Fifty mg of each homogenized sample were transferred into 50 mL glass

vessels. They were incinerated over night at 480˚C until the ash was greyish-white. After cooling down of the ashes, 1 mL of extraction solution (1.5% w/v hydroxylammonium chloride in 6 M HCl) was added to each sample. After 10 minutes of incubation, 9 mL of deionized water were added. The solution was thoroughly shaken and filtered through a blue band filter. The following elements were analyzed: zinc, boron, barium, strontium, aluminum, iron, manganese, calcium, magnesium, phosphorus, sodium and potassium. The measurements took place on an inductively coupled plasma optical emission spectrometer (ICP-OES; Varian 725-ES, Agilent Technologies, Santa Clara, CA, USA).

Total carbon and nitrogen contents were analyzed in 5–7 mg of the oven-dried and homogenized leaf material after dry combustion on a Vario EL III elemental analyzer (Elementar Analysensysteme, Hanau, Germany). The high organic standard (HOS; IVA Analysentechnik, Meerbusch, Germany) with 7.45% C and 0.52% N was used as a reference (charge 287371/264236).

## Statistical analyses

All statistical analyses were conducted with the statistics software R version 3.5.1 [40] in RStudio version 1.1.456. For experiment 1, the influence of soil texture (soil origin) and soil treatment on shoot length after eight weeks, shoot and root dry mass and their interaction was investigated by a two-way analysis of variance (ANOVA). Differences of normalized gene expression between untreated ARD soil and γ-irradiated ARD soil of the first experiment were analyzed using Student's t-Test for each soil and time point separately. For the second experiment, a linear model was fitted for the expression data of each gene of interest, considering each time point separately. Multiple comparisons were calculated using a Tukey test within the "multcomp" package version 1.4–10 [41]. Shoot lengths were analyzed separately by week. A linear model was fitted and differences between the soil treatments were assessed with a Tukey-Duckworth test for p < 0.05 using the "multcomp" package.

Phytoalexin data were evaluated with all-pairs interaction contrast analysis using sandwich estimator (Tukey test) in R with the packages "sandwich" [42] and "multcomp" [41] and p < 0.05 for the compact letter display. For the data containing multiple zeros (non-detected amounts of phytoalexins), nonparametric multiple comparisons for relative contrast effects were applied within the R package "nparcomp" version 3.0 [43].

Significant differences for the nutrient analysis were evaluated using linear models and least square means in R. The correlation analysis of the relationship between two parameters was performed using linear regression analysis. The Pearson correlation coefficient, r, was calculated using Sigma Plot 11.0 (Systat Software Inc., San Jose, USA). Statistical significance (P-value) was tested using one-way univariate ANOVA with p < 0.05.

## Results

The following section separately describes the results of the two experiments, starting with data of candidate gene identification (experiment 1) and followed by results from candidate gene validation (experiment 2).

### Experiment 1: ARD affected soils influenced growth of apple rootstock M26 to different extents

The three ARD soils tested in this experiment influenced plant growth to different extents, with the sandy soil Heidgraben having the strongest overall impact. After eight weeks, the strongest reduction in shoot growth was visible in soil from Heidgraben (34.9 ± 4.1 cm in γARD to 10.5 ± 5.7 cm in ARD), followed by Ruthe (37.6 ± 4.7 cm in γARD to 27.4 ± 7.4 cm in

ARD) and Ellerhoop (22.7 ± 7.0 cm to 16.7 ± 5.8 cm). In soil from Ellerhoop, shoot length did not differ significantly from week 7 on and individual plants even had a greater shoot length in ARD soil compared to γARD (Fig 1). The ANOVA showed a significant influence of both the soil origin and soil treatment (untreated, disinfected) on shoot length after 8 weeks (p < 0.001 for both). Additionally, shoot length was affected differently by the combination of soil texture and treatment, significantly as seen by a p-value of 1.884e-05 for the interaction of the two parameters.

All ARD variants negatively affected the root system, showing a darker coloration and considerably less branching and biomass (Figs 1 and 2). Browning and blackening of the roots was also visible for the plants showing only little to no reduction in shoot growth. In general, plants grown in Ruthe soil achieved the longest shoots, followed by Heidgraben and Ellerhoop.

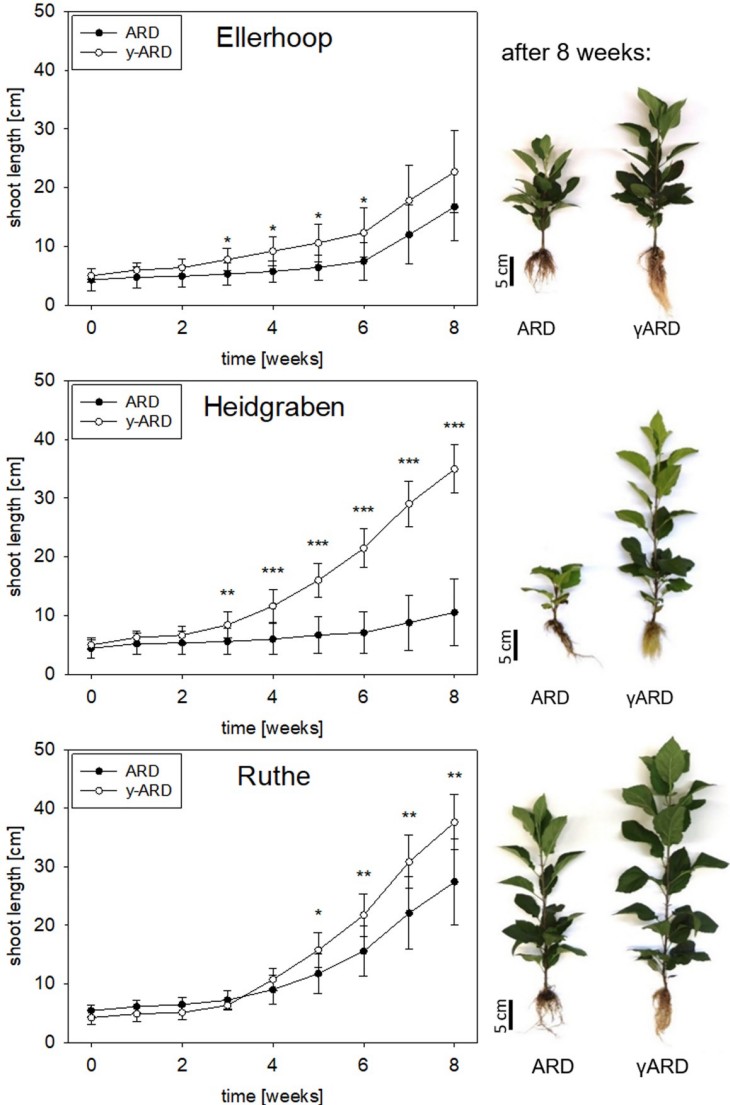

**Fig 1. Growth of M26 plants in untreated ARD soil (ARD) and γ-irradiated ARD soil (γARD) from the three sites Ellerhoop, Heidgraben and Ruthe over eight weeks.** Photos depict representative plants of each variant at the end of the experiment. Means ± SD, n = 10, Tukey-Duckworth test, significant differences at each time point shown for p < 0.05 (*), p < 0.01 (**) and p < 0.001 (***).

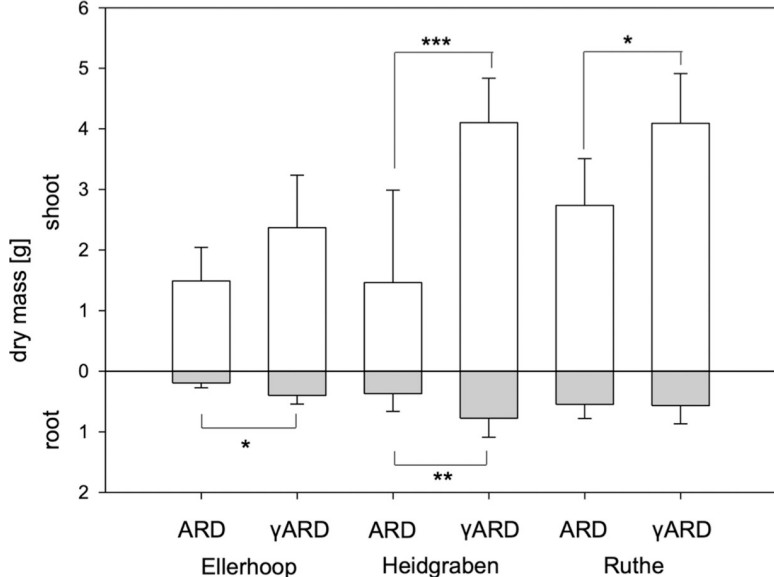

**Fig 2. Shoot and root dry masses of M26 plants after 8 weeks of cultivation in ARD or γARD soil from the three sites Ellerhoop, Heidgraben and Ruthe.** Means ± SD, n = 10, differences between ARD and γARD assessed with Tukey-Duckworth test for each soil (p < 0.05 (*), p < 0.01 (**) and p < 0.001 (***)).

The pattern of shoot dry mass followed that of the shoot length (Fig 2). The strongest reduction in shoot dry mass of ARD *versus* γARD plants was present in Heidgraben, followed by Ruthe and Ellerhoop, where the reduction was not statistically significant after 8 weeks. The root dry masses on the other hand showed a significant reduction in Heidgraben and Ellerhoop soil, but no reduction was observed for plants grown in soil from Ruthe (Fig 2). Both shoot and root dry mass were significantly influenced by both soil and treatment (p < 0.001 for both shoot dry mass and p < 0.001 and p < 0.05 for root dry mass respectively). An interaction between soil texture and disinfection treatment was again present for shoot dry mass with p = 0.012.

## Experiment 1: The transcription factor *ERF1B* and the phytoalexin biosynthesis genes *BIS3* and *B4Hb* showed distinct early differences between γARD and ARD variants

The four potential biomarker genes for early detection of ARD, *1-aminocyclopropane-1-carboxylate oxidase homolog 1-like (ACO1), gibberellin-regulated protein 1-like (GASA1)* and *GATA zinc finger domain-containing protein 10-like* (*GATAD10*) were not distinctly expressed between the γARD and ARD variants in the three ARD soils investigated (S1 Fig). *ACO1* showed a slight trend of higher expression in the γARD variants, which was significant at day 3 in the soil from Ruthe and at day 7 in Ellerhoop soil. In general, the *ACO1* transcript was of low abundance in all variants (S1 Fig). The *GATAD10* transcript of the ARD variant showed a peak on day 1. This was due to only one of the four biological replicates each from Ruthe and Heidgraben showing a considerably higher *GATAD10* expression than the other three replicates, as indicated by the high standard deviation.

The *ethylene-responsive transcription factor 1B-like* (*ERF1B*) showed significant differences in gene expression between the untreated and irradiated variants of all three ARD soils from day 3 on (Fig 3A). In Ellerhoop soil, a slight trend of higher upregulation was observed at day

7 in comparison to Ruthe and Heidgraben, where the expression was rather comparable between day 3 and 7.

A strong and fast gene expression response was found for the two phytoalexin biosynthesis genes *biphenyl synthase 3* (*BIS3*) and *biphenyl 4-hydroxylase* (*B4Hb*). A significant difference in gene expression between ARD and γARD was found for all soils after 3 days with a higher expression of both genes in the ARD variants. *B4Hb* was the only exception in Ellerhoop where the expression differed significantly only after 7 days (Fig 3B and 3C). The fastest response was observed in soil from Ruthe, where the BIS3 transcript reached a plateau already after 3 days. The highest *BIS3* expression was found after 56 days in soil from Ellerhoop. *B4Hb* expression showed similar patterns as *BIS3*, but on a lower overall expression level.

Due to their consistent upregulation in roots affected by ARD, the three genes *ERF1B*, *BIS3* and *B4Hb* were further tested in experiment 2 for their specificity for ARD.

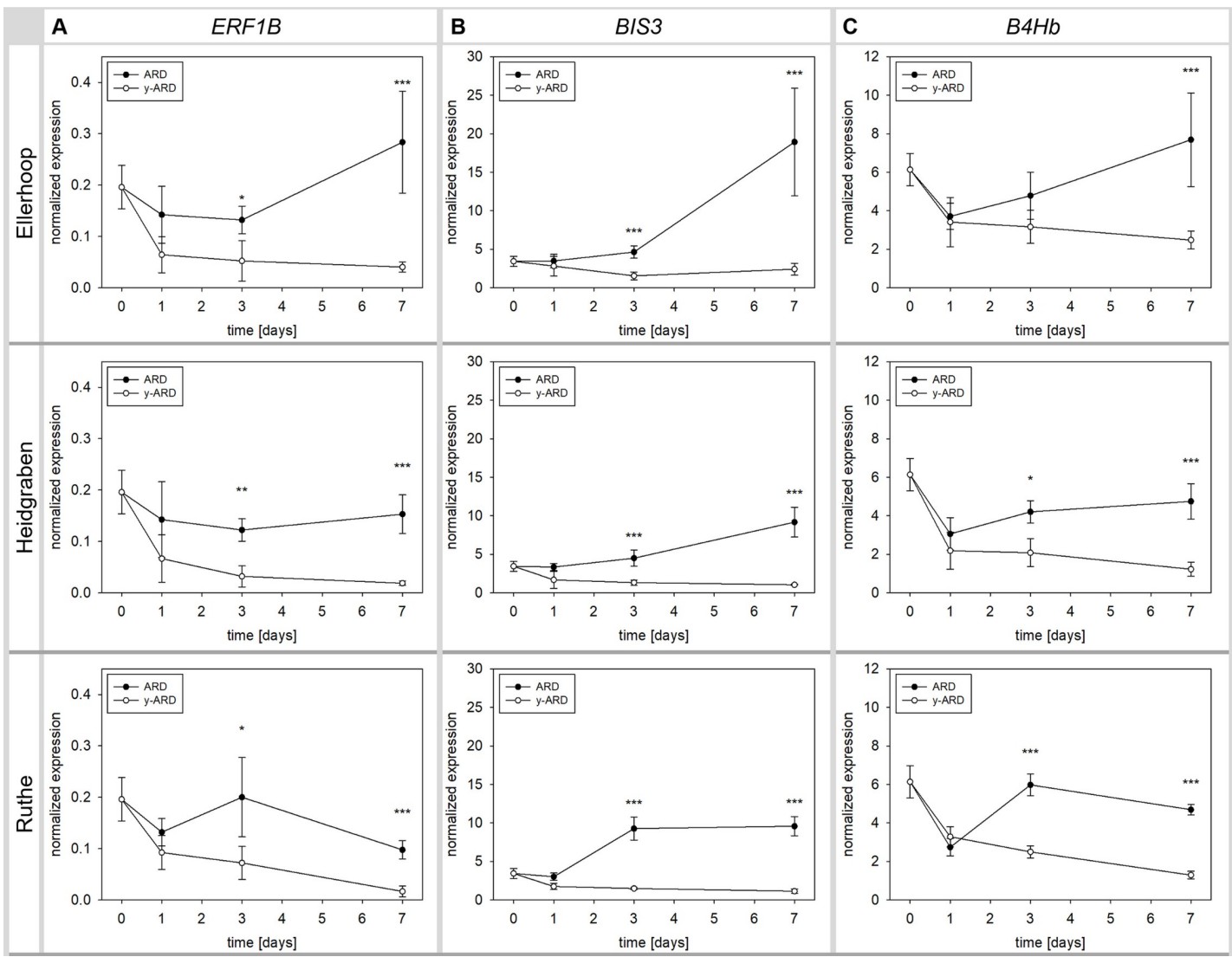

**Fig 3. Normalized gene expression of** *ethylene-responsive transcription factor 1B-like* (*ERF1B*), *biphenyl synthase 3* (*BIS3*) and *biphenyl 4-hydroxylase* (*B4Hb*) in M26 roots growing in ARD and γARD soil from the three sites Ellerhoop, Heidgraben and Ruthe. Means ± SD, n = 4 pooled samples. Significant differences at each time point shown for $p < 0.05$ (*), $p < 0.01$ (**) and $p < 0.001$ (***) as investigated by Tukey tests.

### Experiment 1: Phytoalexin contents were elevated in the root shortly after potting into untreated ARD soil

Total phytoalexin contents in roots grown in ARD soils for 56 days were significantly increased compared to those grown in γARD soils from all three locations (Fig 4). The highest total phytoalexin content was found in roots grown in soil from Heidgraben with an average content of 1.082 mg g$^{-1}$ DW. In all three soils, the total phytoalexin contents increased rapidly during the initial days they were still high at day 56 (S3 Fig and Fig 4). The total phytoalexin content in the roots correlated strongly with shoot dry mass ($p < 0.0001$) and less pronounced with root dry mass ($p < 0.05$).

Among the four biphenyls and four dibenzofurans identified by GC-MS, 2-hydroxy-4-methoxydibenzofuran (RI = 2131) was the only phytoalexin that was detected in all samples including day 0. The average content of 2-hydroxy-4-methoxybiphenyl in roots grown for 56 days on γARD soil was 0.014 mg g$^{-1}$ DW, whereas it reached to 0.657 mg g$^{-1}$ DW in roots grown in ARD soils for 56 days, which corresponds to a 47-fold increase. Also, all other seven phytoalexin compounds were induced in the ARD variants (Fig 4). Noraucuparin (RI = 2121) and noreriobofuran (RI = 2259) were the most strongly induced biphenyl and dibenzofuran compounds, respectively, in roots from ARD soils, with a significant increase in Ellerhoop and Ruthe ARD variants (Fig 4). These individual compounds showed a significant negative correlation with shoot dry mass: noraucuparin ($p = 0.0001$), noreriobofuran ($p < 0.001$), hydorxyeriobofuran ($p < 0.001$), eribofuran ($p < 0.05$) and 2-hydroxy-4-methoxydibenzofuran ($p = 0.0001$). Furthermore, noraucuparin ($p < 0.05$), 2-hydroxy-4-methoxydibenzofuran ($p < 0.01$), and eribofuran ($p < 0.05$) were negatively correlated with the root dry mass.

### Experiment 2: Abiotic stressors influenced plant growth and nutrient composition

For experiment 2, plants were grown under different abiotic stress conditions (heat, salt and nutrient starvation) and in a "healthy" control soil from a grass plot (grass) (Table 1) to further characterize the selected early ARD biomarker candidate genes *ERF1B*, *BIS3 and B4Hb* from experiment 1. Plants showed characteristic symptoms related to the applied stresses. The lack

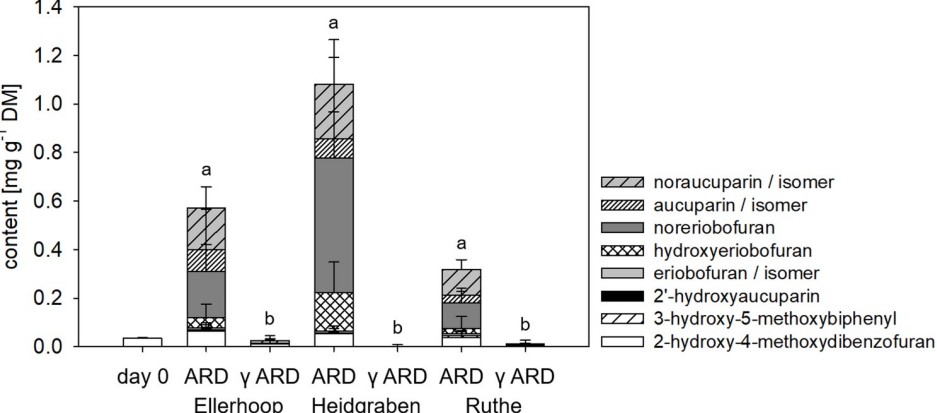

**Fig 4. Means and standard deviations of individual phytoalexin compounds measured in M26 roots grown in ARD or γARD soil from the sites Ellerhoop, Heidgraben and Ruthe after 8 weeks.** n = 7 for Ellerhoop ARD, n = 8 for Ellerhoop γARD, n = 6 for Heidgraben and Ruthe. Different letters indicate significant differences in total phytoalexin content as calculated by multiple comparisons (Tukey test, p < 0.05). The results of the statistical comparisons of the individual compounds are given in Table 3 below.

**Table 3. Results of all-pairs interaction contrast analysis using sandwich estimator of individual phytoalexin compounds measured in M26 roots grown in ARD or γARD soil from the sites Ellerhoop, Heidgraben and Ruthe after 8 weeks (see Fig 4).** Different letters indicate significant differences in content as calculated by multiple comparisons (all-pairs interaction contrast analysis using sandwich estimator or nonparametric multiple comparisons for relative contrast effects, p < 0.05).

| Phytoalexin compound | Ellerhoop | | Heidgraben | | Ruthe | |
|---|---|---|---|---|---|---|
| | ARD | γARD | ARD | γARD | ARD | γARD |
| 2-hydroxy-4-methoxydibenzofuran | a | bc | ab | c | ac | c |
| 3-hydroxy-5-methoxybiphenyl | a | a | a | a | a | a |
| 2'-hydroxyaucuparin | a | a | a | a | a | a |
| eriobofuran | a | a | a | a | a | a |
| hydroxyeriobofuran | ab | ab | a | b | ab | b |
| noreriobofuran | a | b | ab | b | a | b |
| aucuparin | a | a | a | a | a | a |
| noraucuparin | a | b | ab | b | a | b |

of nutrients especially inhibited both shoot and root biomass in the nutrient starvation variant and led to stunting and leaf chlorosis (Fig 5 and S4 Fig). The oldest leaves of the heat stress variant showed light brown angular necrotic spots as heat stress symptoms (S4 Fig). When the plants were transferred to the greenhouse, newly grown leaves did not show these specific symptoms anymore. The plants subjected to salt stress showed symptoms of leaf margin necrosis at the oldest leaves (S4 Fig). As watering was continued with regular tap water, the newly grown leaves did not show such symptoms anymore. Plants grown in ARD soil showed shortened internodes and darkened roots. In comparison to the variants grown in peat substrate, both variants grown in soil, ARD and grass, showed a browning of roots and reduced growth (Fig 5 and S4 and S5 Figs). Growth inhibition by the abiotic stresses was also reflected in a significant reduction in both shoot and root dry mass after 8 weeks (Fig 5).

A nutrient analysis of the plants from experiment 2 was conducted to test if (i) the nutrient starvation variant indeed caused deficient levels of nutrients in the plants and (ii) a possible connection between candidate gene expression and general nutrient status exists. All variants except the nutrient starvation variant were fertilized with Osmocote Exact 3–4 M (Table 1). Consequently, the nutrient starvation variant had significantly smaller contents of nitrogen (7.71 mg g$^{-1}$ DM) and phosphorous (0.81 mg g$^{-1}$), which were approximately half the content of the other variants (Table 4). For potassium, however, contents were smaller in all non-soil

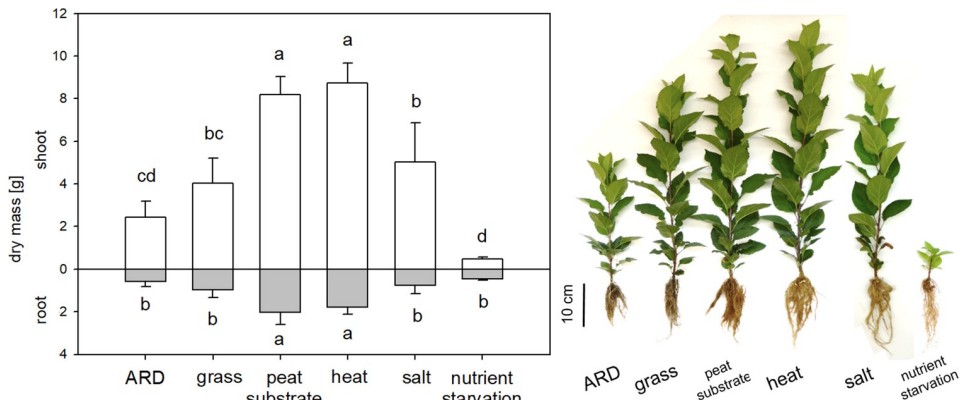

**Fig 5. Shoot and root dry masses of M26 plants grown in the variants ARD soil, grass soil, peat substrate, heat, salt stress and nutrient starvation after 8 weeks (n = 5, except for salt stress n = 4).** Different letters indicate a statistical difference (Tukey Test, p < 0.05) between variants. See Table 1 for details on the variants.

**Table 4. Nutrient contents [mg g$^{-1}$ dry mass] in leaf material of M26 plants grown under the conditions ARD, grass soil, peat substrate, heat, salt stress and nutrient starvation as depicted in Table 1.** Different lower case letters indicate significant differences between treatments (tested by estimated marginal means, p = 0.05), (means ± SD of n = 5, except for salt stress n = 4).

| Content [mg g$^{-1}$ DM] | ARD | Grass | Peat substrate | Heat | Salt | Nutrient starvation |
|---|---|---|---|---|---|---|
| Al | 0.22 ± 0.05a | 0.3 ± 0.06a | 0.26 ± 0.04a | 0.24 ± 0.03a | 0.25 ± 0.04a | 0.31 ± 0.08a |
| B | 0.2 ± 0.06ab | 0.27 ± 0.05a | 0.25 ± 0.05a | 0.21 ± 0.03ab | 0.21 ± 0.02ab | 0.15 ± 0.01b |
| Ba | 0.16 ± 0.04ab | 0.21 ± 0.05a | 0.17 ± 0.03ab | 0.16 ± 0.02ab | 0.16 ± 0.02ab | 0.12 ± 0.01b |
| C | 480.63 ± 3.15a | 478.97 ± 2.72a | 477.35 ± 2.8a | 478.37 ± 3.71a | 477.41 ± 3.53a | 459.33 ± 4.2a |
| Ca | 7.58 ± 0.64a | 7.18 ± 0.92a | 8.87 ± 0.84a | 8.1 ± 0.22a | 8.43 ± 0.93a | 13.39 ± 1.38b |
| Fe | 0.06 ± 0.01a | 0.09 ± 0.01ab | 0.06 ± 0.01a | 0.06 ± 0.01a | 0.06 ± 0.02a | 0.13 ± 0.05b |
| K | 11.93 ± 1.82ab | 12.88 ± 1.71a | 7.36 ± 0.42c | 7.5 ± 2.1c | 9.47 ± 1.35bc | 6.38 ± 0.71c |
| Mg | 2.09 ± 0.18a | 2.17 ± 0.14a | 2 ± 0.25a | 1.94 ± 0.2a | 2.03 ± 0.12a | 1.5 ± 0.19b |
| Mn | 0.05 ± 0a | 0.04 ± 0b | 0.05 ± 0.01a | 0.04 ± 0.01ab | 0.04 ± 0b | 0.04 ± 0b |
| N | 18.14 ± 3.04a | 17.27 ± 2.07ab | 12.73 ± 1.16b | 13.39 ± 2.55b | 14.09 ± 2.7b | 7.71 ± 1.69b |
| Na | 2.23 ± 0.38ab | 3.14 ± 0.38c | 2.87 ± 0.46ac | 2.61 ± 0.4ac | 2.73 ± 0.59ac | 1.59 ± 0.2b |
| P | 1.62 ± 0.16a | 1.88 ± 0.4a | 1.69 ± 0.34a | 1.66 ± 0.12a | 1.8 ± 0.14a | 0.81 ± 0.13b |
| Sr | 0.016 ± 0.002ab | 0.019 ± 0.003a | 0.015 ± 0.001ab | 0.014 ± 0.002b | 0.014 ± 0.002b | 0.037 ± 0.003c |
| Zn | 0.04 ± 0a | 0.04 ± 0a | 0.03 ± 0bc | 0.03 ± 0.01bc | 0.02 ± 0c | 0.04 ± 0.01ab |

based variants with 6.38 mg g$^{-1}$ DM in nutrient starvation, 7.5 mg g$^{-1}$ DM in heat stress, 9.47 mg g$^{-1}$ DM in salt, and 7.36 mg g$^{-1}$ DM in the peat substrate variant. Interestingly, the calcium content in the nutrient starvation variant was almost twice as high as in the other variants (Table 4).

## Experiment 2: Expression of *BIS3* and *B4Hb* increased in ARD affected roots from week 1 to week 2, but was barely influenced by abiotic stressors

Gene expression of the two early ARD biomarker candidate genes belonging to the phytoalexin biosynthesis pathway, *biphenyl synthase 3* (*BIS3*) and *biphenyl 4-hydroxylase b* (*B4Hb*) showed a significantly higher normalized expression level in ARD soil compared to the abiotic stress conditions investigated (Fig 6A and 6B). Interestingly, a considerable expression of both *BIS3* and *B4Hb* was found in roots of the grass variant. At week 1, normalized expression between the grass and ARD variant was very similar. At week 2, *BIS3* and *B4Hb* expression remained at a similar level in the grass variant but was elevated in the ARD variant. For *BIS3*, the expression level in the heat, salt and nutrient starvation variants was significantly lower than in the ARD and grass variants for both time points. Gene expression in the abiotic stress variants did not differ significantly from the expression of the acclimatized plants (day 0) and the peat substrate variant (Fig 6A). *B4Hb* expression on the other hand was significantly higher in the heat and salt variants compared to the peat substrate variant at week 1, reaching a comparable level to the grass variant but being lower than in roots of the ARD variant (Fig 6B). At week 2, the *B4Hb* expression pattern resembled that of *BIS3*. Overall, these two biosynthetically linked genes followed almost the same expression pattern across all variants and time points, with *BIS3* being the more highly expressed gene.

The expression pattern of the *ethylene-responsive transcription factor 1B* (*ERF1B*) was prone to much higher variation in comparison to the highly expressed phytoalexin biosynthesis genes (Fig 6C). The high variation is likely caused by its low overall expression level. *ERF1B* expression followed a similar pattern as the two other candidate genes with a significant higher expression in the ARD variant at week 2 compared to the other variants. However, for *ERF1B*

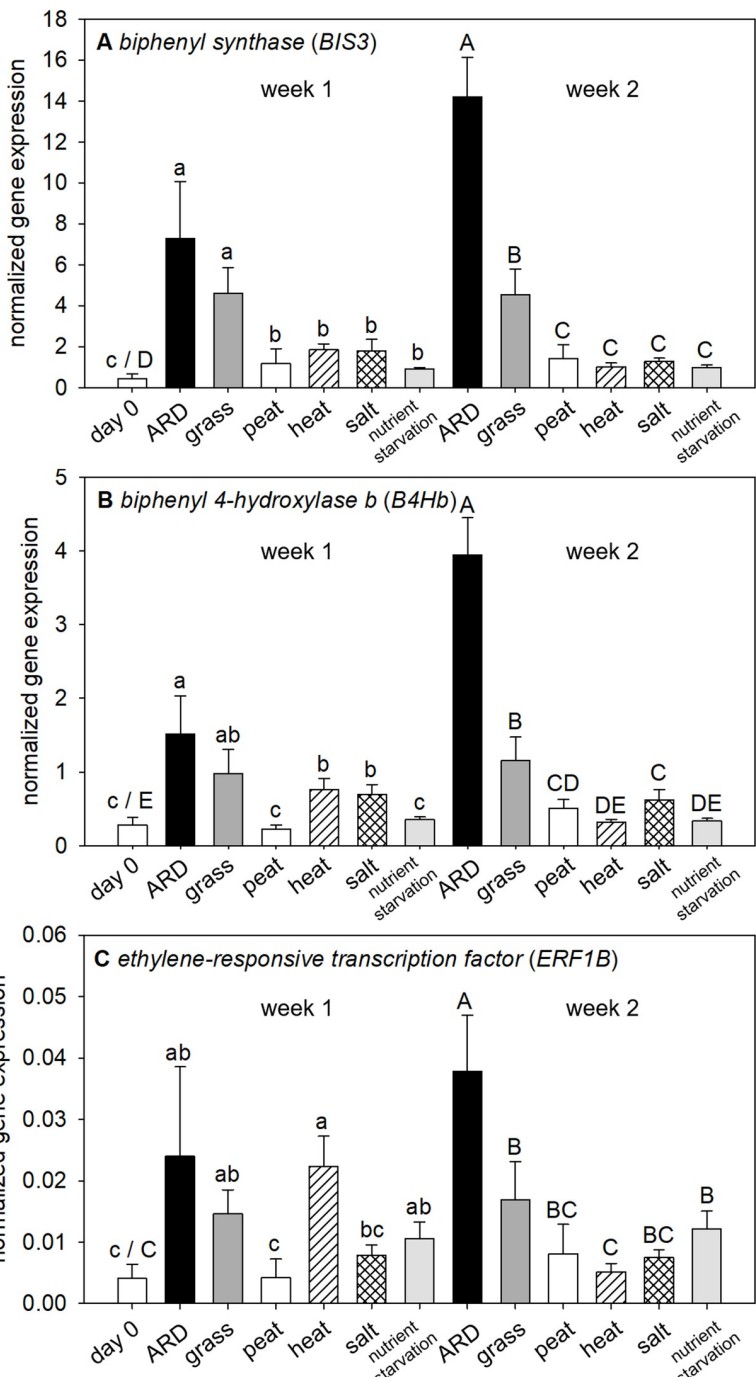

**Fig 6. Normalized gene expression of the biomarker candidate genes** *biphenyl synthase 3* (*BIS3*) **and** *biphenyl 4-hydroxylase* (*B4Hb*) **and** *ethylene-responsive transcription factor 1B-like* (*ERF1B*) **in roots of M26 plants grown under the conditions ARD, grass soil, peat substrate, heat, salt stress and nutrient starvation as depicted in** Table 1. Means ± SD, n = 5, differences between variants were assessed with a Tukey test for each week (p < 0.05). Different letters indicate significant differences between variants for week 1 (small letters) and week 2 (capital letters).

a clear response to heat stress was observed. At week 1, *ERF1B* gene expression showed a clear peak for the heat stressed plants (Fig 6C), which was not present anymore at week 2, after the plants had been relieved from the stress.

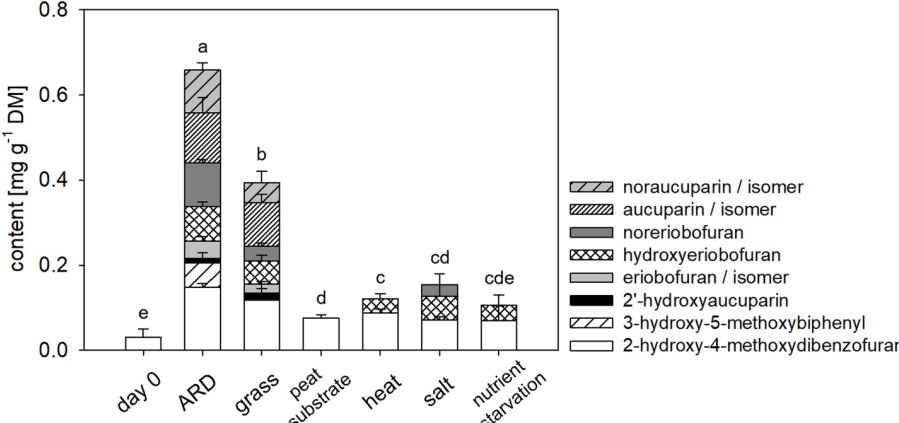

**Fig 7. Means of individual phytoalexin compounds measured in M26 roots grown under the conditions ARD, grass soil, peat substrate, heat, salt stress and nutrient starvation after 14 days of culture as depicted in Table 1.** Means and standard deviations of 5 plants, except for salt stress (n = 4) are given in the table below.

## Phytoalexin composition was most diverse in the ARD variant compared to other stressors

The total phytoalexin content in roots from experiment 2 after two weeks (Fig 7) correlated with not only the expression of the phytoalexin biosynthesis genes *biphenyl synthase 3* (*BIS3*, *p < 0.0001*) and *biphenyl 4-hydroxylase b* (*B4Hb, p < 0.0001*) but also with the *ethylene-responsive transcription factor 1B-like* (*ERF1B*, p < 0.0001) (Fig 6A and 6B). The highest total phytoalexin content was present in the ARD variant, followed by the grass variant. In addition, the composition of individual phytoalexin compounds was more diverse in the soil variants ARD and grass than in the other variants. 2-Hydroxy-4-methoxydibenzofuran was again the only phytoalexin detected at a base level in all variants (Fig 7). Hydroxyeriobofuran was present in the soil variants ARD and grass and in the abiotic stress variants heat, salt and nutrient starvation. Furthermore, the salt variant contained an additional small amount of noreriobofuran (Fig 7), which was significantly smaller compared to the ARD and grass variants. The peat substrate did not induce any new phytoalexin. Roots grown in grass soil contained the same profile of phytoalexins as the ARD variant with only 3-hydroxy-5-methoxybiphenyl being exclusively present in the ARD variant. Although noraucuparin was detected in both ARD and grass variants, it was only significantly induced in the ARD variant as well (Fig 7).

The detected content of all phytoalexin compounds, except for 2'hydroxyaucuparin and hydroxyeriobofuran, strongly correlated (p = < 0.0001) with the gene expression levels of *BIS3*, *B4Hb* and *ERF1B* (see Table 5). The abundance of single phytoalexin compounds was correlated with shoot and root dry mass, e.g. for noraucuparin (shoot p < 0.05, root p < 0.05), eribofuran (shoot p < 0.05, root p < 0.05), noreriobofuran (shoot p < 0.05, root p < 0.05) and hydroxyeriobofuran (shoot p < 0.05, root p < 0.001).

## Discussion

The experiments conducted in this study aimed at the identification and validation of universal and early biomarkers for ARD responses within the apple roots on the gene expression level. The first experiment compared the reaction of M26 apple plants growing in soils from three ARD sites to identify genes that are universally expressed under ARD conditions. The second experiment investigated the responses of three selected biomarkers to different abiotic stressors to exclude general stress response genes.

**Table 5. Results of all-pairs interaction contrast analysis using sandwich estimator of individual phytoalexin compounds measured in M26 roots grown under the conditions ARD, grass soil, peat substrate, heat, salt stress and nutrient starvation after 14 days of culture (Table 1, Fig 7).** Different letters indicate significant differences in content as calculated by multiple comparisons (all-pairs interaction contrast analysis using sandwich estimator or nonparametric multiple comparisons for relative contrast effects, p < 0.05).

| Phytoalexin compound | day 0 | ARD | grass | peat substrate | heat | salt | nutrient starvation |
|---|---|---|---|---|---|---|---|
| 2-hydroxy-4-methoxydibenzofuran | d | a | ab | bc | bc | c | bcd |
| 3-hydroxy-5-methoxybiphenyl | b | a | b | b | b | b | b |
| 2'-hydroxyaucuparin | a | a | a | a | a | a | a |
| eriobofuran | b | a | a | b | b | b | b |
| hydroxeriobofuran | c | a | ab | c | b | abc | bc |
| noreriobofuran | c | a | b | c | c | bc | c |
| aucuparin | b | a | a | b | b | b | b |
| noraucuparin | b | a | b | b | b | b | b |

## Differences in shoot growth alone may not be suitable to assess ARD severity in a soil

With ARD coming to the awareness of fruit and tree producers, methods of identifying and quantifying the disease in the soil are more than ever of interest. Yim et al. [10] developed a bio-test by means of linking the shoot length reduction of the highly ARD-susceptible rootstock M26 in ARD-affected soils compared to the respective disinfected soils as an indication for the presence of ARD. We used this bio-test to compare the reaction of M26 when grown in soil from three ARD sites in a greenhouse experiment. At these sites (Ellerhoop, Heidgraben and Ruthe), ARD had been intentionally induced by repeatedly replanting the apple rootstock 'Bittenfelder Sämling' [12]. The ARD sites share the same cropping history, but differ in their soil structural/physical and chemical parameters and were exposed to different climatic conditions.

After growing in the soils for eight weeks, M26 plants showed typical ARD symptoms in comparison to the respective γ-disinfected controls (Fig 1). The roots of the plants grown in the untreated ARD soil were dark in coloration, which has been reported to be due to cellular damage and necrosis [1, 10, 44] but also deposition of phenolic compounds [10]. In addition, the plants of the untreated ARD variants in all soils had fewer fine roots, which resulted in a significantly reduced root dry mass in ARD soil from Ellerhoop and Heidgraben (Fig 2). A reduction in shoot length and biomass, another major feature of ARD (reviewed in [1]), was observed for soils from Ruthe and Heidgraben (Figs 1 and 2). In the Ellerhoop soil, shoot length and shoot dry mass did not significantly differ between the ARD and γARD variants after eight weeks. This could point to the fact that either chemical or physical characteristics of this soil caused slower growth, beneficial soil organisms were lacking or pathogens survived the irradiation and thus occurred in the γ-irradiated soil. The latter speculation is supported by the observation that roots grown in γ-irradiated Ellerhoop soil contained significantly higher phytoalexin contents compared to the γ-irradiated soils of the other two sites investigated (Fig 4).

The strongest ARD effect in terms of shoot length and fresh mass reduction was found for the sandy soil from Heidgraben (Figs 1 and 2). It has been reported that ARD is easier and faster induced in sandy soils [45]. Ruthe, as a considerably clayey soil, might have had a greater ARD suppressing capacity as reviewed for general disease suppressiveness by Garbeva et al. [45]. The different water holding capacities of the sandy and loamy soil could have also contributed to the observed differences. The lower ARD incidence in this experiment in Ruthe as well as Ellerhoop soil may also be due to waterlogging after heavy rain falls at the experimental

sites. Anaerobic conditions in the soil caused by waterlogging may have an effect similar to other disinfecting measures leading to a weakening of the ARD effect [28, 46, 47].

With only information on shoot length differences from the bio-test, the Ellerhoop soil would have been classified as not or only weakly affected by ARD. It is likely that factors apart from ARD influenced plant growth in this variant. As the precise cropping history of the site Ellerhoop is known, the presence of a replant situation can here be confirmed. However, if the precise history is unknown, as it is the case for most commercial sites, shoot growth reduction alone might not be sufficient to test for the presence of ARD. Other typical ARD symptoms could be observed in the Ellerhoop ARD variants such as root discoloration and reduction in fine roots. A microscopic evaluation of root symptoms has been proposed as a measure to evaluate the presence and severity of ARD early on [44]. An advantage of this analysis is an early-on diagnosis, but routine and experience are required for a microscopic evaluation and the analysis of a large number of samples is time consuming. Therefore, alternative measures for early ARD diagnosis are needed and were identified in the marker genes described below.

### *BIS3* and *B4Hb* are promising candidates for early ARD indication

Finding an early indicator to assist in the diagnosis of ARD is of major importance. The same importance is attributed to a better understanding of the molecular etiology of ARD in order to deduce effective strategies against it, as the present counteractions present only temporal solutions and are uneconomic and not sustainable (reviewed in [9]). The selection of differently regulated genes from Weiß et al. [11, 33] was narrowed down by testing their expression on further ARD soils in this study. *ACO1*, *GASA1* and *GATAD10* did not show the same difference in regulation between the untreated and γARD variants as reported by Weiß et al. [33], although there was a slight consensus in the expression patterns over time (S1 Fig). The discrepancy between the results obtained in this experiment and the results from Weiß et al. [11, 33] may be caused by other environmental factors during the experiment, which could have triggered the regulation of these genes. As experiment 1 took place in winter (February and March 2017) and Weiß et al. conducted their experiment in March 2014 [11] and September 2014 [33], differences in light and temperature as well as pathogen pressure due to the time of year are expected. Further differences in plant growth and molecular response are anticipated due to the usage of different soils, connected with different cropping histories and inherent microbiome.

In contrast, the remaining three genes, *ERF1B*, *BIS3* and *B4Hb*, are promising candidates for a linkage with ARD and therefore as potential biomarkers for ARD. Taken the presented results into account, the expression of these genes has been shown to differ in response to ARD in three different experiments and in five ARD soils of different origin (this study and [11, 33]). Analyzing these three candidate genes in apple roots in response to the abiotic stressors heat and salt stress and nutrient starvation yielded different specificities, which are presented in detail below.

Expression of the *ERF1B* transcription factor was lower than that of the enzyme-coding genes *BIS 3* and *B4Hb*, resulting in a higher variation in qPCR measurements. Its expression roughly followed the phytoalexin biosynthesis genes with the exception of a clear induction upon heat stress (Fig 6C). At week 2, expression in the heat treated plants resumed to a level comparable to the peat substrate without additional stress. The heat stress was removed after the week 1 sampling time point, indicating that the heat stress induction of *ERF1B* gene expression was temporarily limited to the incidence of stress and declined after that.

Ethylene response factors (ERF) are integral components of ethylene signaling and response. They play a major regulatory role in the molecular response to biotic stressors. Upon

activation, they bind to GCC box elements [48]. The ERF1B transcription factor may link plant hormone signaling pathways of ethylene, jasmonic acid and salicylic acid [49, 50] and is involved in biotic stress responses [49, 51, 52]. Constitutive expression of *ERF1B* has been reported to increase resistance against the necrotrophic fungi *Botrytis cinerea* and *Plecto-sphaerella cucumerina* in *Arabidopsis thaliana* [49]. In our study, *ERF1B* expression showed a clear induction in ARD soil (Figs 3 and 6C). *ERF1B* expression can be related to the presence of biotic agents in the soil, which are the main causal agents of ARD [5, 10, 53]. As demonstrated in this study, *ERF1B* expression in apple was additionally linked to heat stress. ERF transcription factors are next to biotic stress responses heavily involved in plant responses to abiotic stresses via binding to dehydration-response elements [54]. Abiotic stresses that induce ERF transcription factor expression include salt stress, drought, cold, heat and changes in light availability (reviewed by [48]). In apple, especially drought and salt stress responses are co-regulated [55, 56]. However, an expression induction of this specific *ERF1B* upon salt stress was not observed in our study (Fig 6C).

*BIS3* and *B4Hb* are coding for enzymes catalyzing subsequent steps in phytoalexin biosynthesis [33]. Phytoalexins are part of the biotic stress response and have been described in members of the *Rosaceae* including *Malus*, *Pyrus* and *Sorbus* [34, 57, 58]. Evidence of their presence in apple roots was first reported by Weiß et al. [11, 33] and in this study. Expression of *BIS3* and *B4Hb* followed a similar expression pattern upon exposure to ARD (Fig 3). They showed an early significant increase on all soils tested as early as three days after the plants were introduced to the ARD soils, with *B4Hb* in Ellerhoop soil as the only exception. Evaluating *BIS3* and *B4Hb* responses to abiotic stressors indicated that they were outstandingly highly expressed in roots of the ARD variant (Fig 6A and 6B). In addition, they were more highly expressed in the grass soil variant (Fig 6A and 6B). This unspecific response is important to consider regarding the limitations of the explanatory power as expression markers but may also give further insights into the ARD etiology as discussed below. The other stresses investigated did not lead to significant induction of these phytoalexin biosynthesis genes. In the literature, phytoalexins are described to be a part of the induced defense response against biotic stressors, fungi in particular [59]. *BIS3* expression was reported previously in apple upon fire blight infection [60] and upon infection with *Pythium ultimum*, one causal agent of ARD [61, 62]. Taking these studies together with further studies from our group [11, 32, 33, 63], the phytoalexin biosynthesis is a clear and highly specific part of the *Malus* response to ARD.

The expression of *BIS3* and *B4Hb* strongly correlated with the abundance of phytoalexin compounds. The maximum total phytoalexin amounts found in this study, reaching up to 1.08 mg g$^{-1}$ dry mass in the ARD variant of Heidgraben in experiment 1 (Fig 4) were similar to maximum amounts of 1.7 mg g$^{-1}$ dry mass after 14 days reported by Weiß et al. [33]. The authors hypothesized about a possible autotoxic effect of the phytoalexin compounds [33], because comparable concentrations of the phytoalexin compounds camalexin and phaseolin were shown to be cytotoxic to *Arabidopsis thaliana*, *Phaseolus vulgaris* and *Beta vulgaris* cells, respectively [64, 65]. In the second experiment, where phytoalexins were detected after 14 days, the measured contents were lower. The highest total content was 0.66 mg g$^{-1}$ dry mass in the ARD variant (Fig 7). One explanation may be the shorter exposure of the plants to ARD in experiment 2 (14 days) compared to experiment 1 (56 days). However, the amounts detected in the study of Weiß et al. [33] were considerably larger after 14 days. This may indicate further factors influencing the total amount of phytoalexins produced.

Individual phytoalexin compounds differed considerably between variants of both experiments. In experiments 1 and 2, 2-hydroxy-4-methoxydibenzofuran was present in all variants and contributed to the total phytoalexin amount detected in the γARD variants (Fig 4) and considerably to the stress variants (Fig 7). Thus, 2-hydroxy-4-methoxydibenzofuran may not

only be considered as a phytoalexin in apple roots, which by definition is synthesized de novo as a (biotic) stress response. As this compound is already present in the plant before stress application, it may be referred to as phytoanticipin, as suggested by van Etten [66].

The phytoanticipin/phytoalexin 2-hydroxy-4-methoxydibenzofuran was present in all the tested variants and contributed considerably to the total phytoalexin amount detected in the γARD variants (Fig 4) and stress variants (Fig 7). When the apple plants were grown in ARD soil, more phytoalexins including four biphenyls and three dibenzofurans were induced. These changes in the phytoalexin profiles were strongly correlated with the expression pattern of the phytoalexin biosynthetic genes (Figs 3 and 6). Of particular interest is 3-hydroxy-5-methoxybiphenyl, which was only detected in the ARD variant of experiment 2 and at day 10 of experiment 1 (S2 Fig), but was absent in all abiotic stresses as well as in grass soil (Fig 7). Apart from 3-hydroxy-5-methoxybiphenyl, the phytoalexin composition of the ARD and grass soils was almost identical. BIS3 is located at the beginning of the biphenyl biosynthesis pathway, while B4Hb converts 3-hydroxy-5-methoxybiphenyl to noraucuparin [33]. The accumulation of 3-hydroxy-5-methoxybiphenyl at earlier time points of ARD indicates a stronger or faster induction of BIS3 than of B4Hb in ARD soil. In contrast, the induction of BIS3 and B4Hb in grass soil was weaker and equal, therefore no accumulation of 3-hydroxy-5-methoxybiphenyl was detected. To improve our understanding of ARD, the role of particularly 3-hydroxy-5-methoxybiphenyl and the enzyme catalyzing its biosynthesis, caffeic acid 3-O-methyltransferase (SaOMT1), may be of interest. In apple, it was shown that *OMTa* expression showed a significant increase in its transcript level, following the same pattern as *BIS3* and *B4Hb* [33].

The nutrient starvation variant showed the smallest contents in all nutrients, especially of N, P and K. An exception was calcium, which had twice the contents than the other variants. Thus, in this nutrient starvation variant, the visual symptoms of nutrient deficiency are proved. Demidchik et al. [67] showed that a decrease of extracellular K leads to a hyperpolarization of the epidermal plasma membrane followed by an increase of cytosolic Ca in *Arabidopsis thaliana*. This would explain the high calcium content in shoot material of the nutrient starvation variant. Based on the gene expression study and phytoalexin concentration inside the root, it can be concluded that abiotic stress has no effect on phytoalexin synthesis and only biotic factors affect the expression of these biomarker genes.

Further studies including soils from more sites should follow to shed additional light on the quantitative correlation between ARD severity and candidate gene expression.

## Buildup of the phytoalexin defense reaction as possible cause of ARD

Results from both experiments presented in this study indicate a time effect in candidate gene expression, with gene expression increasing in the ARD variants (Figs 3 and 6 and S3 Fig). Weiß et al. [33] investigated gene expression in apple roots grown 3, 7, 10 and 14 days in ARD versus γARD soil and observed patterns similar to those we report in this study. With the comparison to grass soil, we have now proven that pathogens present in this control soil also induce the expression of our candidate genes *BIS3*, *B4Hb* and *ERF1B* to a certain extent (Fig 6). Candidate gene expression in ARD soil was complemented by the phytoalexin analysis. After one week, the expression of all three genes in grass soil was nearly comparable to the one in ARD soil, but after two weeks there was a significantly higher expression in ARD soil. Previous studies also reported growth reduction and browning of roots in grass soil in comparison to γ-irradiated grass or ARD soil [12, 44]. Thus, we assume a molecular defense reaction against biotic stress to be also initiated in grass soil, but in ARD soil it was becoming much stronger with time.

ARD develops in healthy soil and consequently an interaction of the plant with the soil leads to the formation of a replant situation. This plant-soil-interaction must be quite unique

to *Malus* and closely-related rosaceous species, as ARD is not affecting non-related species outside the family. Winkelmann et al. [9] proposed the cause of ARD to be harmful shifts in soil microbiota (dysbiosis) regarding their structure as well as their functions. Radl et al. [68] attributed these harmful shifts to the previous culture of the same or closely-related species. As biphenyls and dibenzofurans are defense compounds specific to the subtribe *Malinae* and produced in lower amounts in healthy soils, one could speculate on their involvement in the shift of soil microbiota leading to the development of ARD. This hypothesis would require ARD agents to survive or even utilize these defense compounds where other microorganisms cannot. Further studies are required to test this hypothesis. One of the promising strategies to prove this hypothesis could be the isolation and identification of microbes from ARD soils and then to investigate the effects of biphenyls and dibenzofuran phytoalexin on these isolated microbes. In case the phytoalexins could serve as carbon source to support or even promote the growth of specific microbes, the increased microbes will further induce the phytoalexin production in roots, which could then cause the damage of root cells by cytotoxic properties of phytoalexins. However, this hypothesis needs to be addressed in future research.

## Conclusions

In the present study, we identified the expression of the phytoalexin biosynthesis genes *BIS3* and *B4Hb* as suitable biomarkers for apple replant disease (ARD). Their expression was strongly and consistently induced in roots of apple plants grown in three different soils affected by ARD compared to γ-irradiated ARD soil. Furthermore, their expression was not influenced by common abiotic stresses. The expression of *BIS3* and *B4Hb* was strongly correlated with the abundance of phytoalexin compounds. Hence, due to their high expression levels and their high specificity, *BIS3* and *B4Hb* can be recommended as early ARD markers. Future studies should include expression analyses of the ARD marker genes after inoculation with causal agents that are part of the ARD complex. Furthermore, it would be interesting to correlate gene expression levels to the microbial community composition in the respective ARD soils. In ARD soil, phytoalexin biosynthesis was increased in comparison to healthy soil. The presence of these apple-specific defensive compounds in healthy soil led to the hypothesis that they might play a role in the ARD etiology by attracting specific communities of soil-borne pathogens.

## Supporting information

**S1 Fig. Normalized gene expression of *1-aminocyclopropane-1-carboxylate oxidase homolog 1-like* (*ACO1*), *gibberellin-regulated protein 1-like* (*GASA1*) and *GATA zinc finger domain-containing protein 10-like* (*GATAD10*) in M26 roots growing in ARD and γARD soil from the sites Ellerhoop, Heidgraben, and Ruthe.** Means ± SD, n = 4 pooled samples. Significant differences at each time point shown for $p < 0.05$ (*), $p < 0.01$ (**) and $p < 0.001$ (***) as investigated by Tukey tests.
(PNG)

**S2 Fig. Means and standard deviations of individual phytoalexin compounds measured in M26 roots grown in either ARD or γARD soils from the sites Ellerhoop, Heidgraben and Ruthe after 10 and 56 days.** Number of root samples analyzed indicated by n.
(PNG)

**S3 Fig. Means and standard deviations of summarized phytoalexin compounds measured in M26 roots grown in either ARD or γARD soils from the sites Ellerhoop, Heidgraben**

**and Ruthe over a course of 56 days.** Number of samples analyzed is indicated in S4 Table.
(PNG)

**S4 Fig. Typical leaf and root symptoms of the variants in experiment 2.** See Table 1 for
details on the variants.
(PNG)

**S5 Fig. Shoot length of M26 grown under the conditions ARD, grass soil, peat substrate,
heat, salt stress and nutrient starvation (n = 5 individual plants, except nutrient starvation
n = 4).** Different letters indicate a statistical difference (Tukey Test, $p < 0.05$) between variants.
See Table 1 for details on the variants.
(PNG)

**S1 Table. Sampling schedule for experiments 1 and 2.**
(XLSX)

**S2 Table. Information on biological and technical replicates of experiment 1.**
(DOCX)

**S3 Table. Information on biological and technical replicates of experiment 2.**
(DOCX)

**S4 Table. Pooling and masses of freeze-dried root material from experiment 1 for phyto-
alexin analysis.**
(DOCX)

## Acknowledgments

We would like to thank our student helpers Jenny Horn and Alaina van Slooten, as well as our
technical staff and gardeners for their support in the realization of the experiments. We thank
Ludwig Hothorn for his expertise in the statistical evaluation of the phytoalexin data. Felix
Mahnkopp-Dirks' and Stefan Weiß' theoretical and practical assistance in carrying out the
experiments is greatly appreciated.

## Author Contributions

**Conceptualization:** Annmarie-Deetja Rohr, Traud Winkelmann.

**Formal analysis:** Annmarie-Deetja Rohr, Jessica Schimmel, Benye Liu.

**Funding acquisition:** Ludger Beerhues, Georg Guggenberger, Traud Winkelmann.

**Investigation:** Annmarie-Deetja Rohr, Jessica Schimmel, Benye Liu.

**Supervision:** Georg Guggenberger, Traud Winkelmann.

**Visualization:** Annmarie-Deetja Rohr, Jessica Schimmel.

**Writing – original draft:** Annmarie-Deetja Rohr, Jessica Schimmel, Benye Liu, Georg Gug-
genberger, Traud Winkelmann.

**Writing – review & editing:** Annmarie-Deetja Rohr, Jessica Schimmel, Benye Liu, Ludger
Beerhues, Georg Guggenberger, Traud Winkelmann.

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
