## [Decision Letter · Decision Letter 0]

8 Jun 2020

PONE-D-20-08054

Identification and validation of early genetic biomarkers for apple replant disease

PLOS ONE

Dear Dr. Rohr,

Thank you for submitting your manuscript to PLOS ONE. After careful consideration, we feel that it has merit but does not fully meet PLOS ONE’s publication criteria as it currently stands. Therefore, we invite you to submit a revised version of the manuscript that addresses the points raised during the review process.

 The study has interesting findings on the differential expression of the two genes (BIS3 and B4Hb) involved in the biosynthesis of phytoalexin in the root under diseased and disease-free conditions. Reviewers have raised queries on the technical aspects and presentation of the manuscript, which need to be carefully looked into and addressed.

We look forward to receiving your revised manuscript.

Kind regards,

Suprasanna Penna

Academic Editor

PLOS ONE

Additional Editor Comments:

The study has interesting findings on the differential expression of the two genes (BIS3 and B4Hb) involved in the biosynthesis of phytoalexin in the root under diseased and disease-free conditions. Reviewers have raised queries on the technical aspects and presentation of the manuscript, which need to be carefully looked into and addressed.

4. We note you have included a table to which you do not refer in the text of your manuscript. Please ensure that you refer to Table 3 in your text; if accepted, production will need this reference to link the reader to the Table.

Reviewers' comments:

Reviewer's Responses to Questions

**Comments to the Author**

1. Is the manuscript technically sound, and do the data support the conclusions?

Reviewer #1: Yes

Reviewer #2: Yes

Reviewer #3: Partly

2. Has the statistical analysis been performed appropriately and rigorously? 

Reviewer #1: Yes

Reviewer #2: Yes

Reviewer #3: I Don't Know

3. Have the authors made all data underlying the findings in their manuscript fully available?

Reviewer #1: Yes

Reviewer #2: Yes

Reviewer #3: Yes

4. Is the manuscript presented in an intelligible fashion and written in standard English?

Reviewer #1: Yes

Reviewer #2: Yes

Reviewer #3: No

5. Review Comments to the Author

Reviewer #1: The submitted manuscript described a carefully designed experiment aiming to reveal the quantified physiological impacts of apple plant growth by ARD pathogen in soils and identified the expression of the phytoalexin biosynthesis genes BIS3 and B4Hb as potentially suitable biomarkers for apple replant disease (ARD). Although obvious variations between soils from different sites are expected, the clear indication of negative effect of these soils to apple plant growth, expressed in plant height and dry weight, is a valuable addition to our understanding of this disease. The identified biomarkers are encouraging for diagnostic purpose to potentially suggest the soil “sickness” level. It can be practically useful to farmer, though the actual effective measure for controlling this disease will still be needed likely from the resistant rootstocks giving the soilborne nature and lack of control measures. Experiment 2 provided evidences to distinguish the potential effective biomarker (such as ERF) under biotic factor and abiotic pressure. Metabolite analysis and identification of phenolic compound certainly added the weight to the study and justified the selection related genes as marker. Overall, the reported results added experimental evidences to understand this disease and generated a potential useful tool to diagnose and quantified the potential impact of ARD soils from individual orchards.

With a few minor issues needed to be resolved, this manuscript is recommended for being accepted for publication.

Here are a few minor issues need to be clarified:

Ln 128-129: “In the grassland plots within the reference sites, no members of the Rosaceae had been grown, thus, this soil served as a control soil not affected by replant disease”. Need to be cautious that some members in this pathogen complex are not specific to ARD, rather they are the wide-spectrum opportunistic necrotrophic attackers. That is why you observed the induction of the target gene at low level.

ln 166-167: “Cultivation during the first three days was carried out without additional lighting. From then on, additional light was provided by SON-T Philips Master Agro 400 W lamps”. Why extra light is required? Is this a random event or special design?

Ln 306-307: “In soil from Ellerhoop, shoot length did not differ significantly from week 7 on and individual plants even had a greater shoot length in ARD soil compared to γARD.” What is the speculation about this observation?

Ln 372-373: “In soils from Ellerhoop and Ruthe, the total phytoalexin contents increased rapidly during day 7 and day 10, decreased at day 56…” what is the speculation of the decrease?

In Figure 4, any comment or speculation on why the second site induce the most increase of secondary metabolite production?

In Fig 6, the expression of these genes was obviously higher in Control soil Heidgraben (grass), obviously higher than “Peat substrate + quartz sand”, does this mean other soil microorganisms, which may not be pathogenic to apple root, triggered the induction of these genes? If this is the case, then ARD pathogens are not the only microorganisms induce the gene expression, rather many other soilborne necrotrophs has the same effect on the expression of these genes.

Ln 528-529: “This could point to the fact that pathogens that affect plant growth were also present in the γ-irradiated soil.” Is it possible this treatment does not eliminate all the pathogenic factors? or alternatively sometimes a growth promoting microorganism existed in ARD soils too.

Reviewer #2: A meaningful work on ARD (Apple replant disease), overall the results were quite convincing as shown by some of the results in the study, but there are stll some minor defects . Here are some of key points.

1.Line 356-357, the response speed of soil in Heidgraben was not the fastest according to your supporting evidences. The expression is vague, BIS3 expression reached a plateau after 3 days, in Ruthe or all?

2.As author suggested, BIS1 and B4Hb can act as early biomarkers for apple replant disease and ineed significantly distinguish between ARD and disinfected ARD. However, whether they can assess different levels of ARD severity or broad-spectrum soils, not limited to just two conditions, ARD with or without. Besides, three kinds of soils may not be universal enough.

3.The resolution of pictures in Fig1 and fig3 are too low.

Reviewer #3: • This study shows the differential expression of the two genes (BIS3 and B4Hb) involved in the biosynthesis of phytoalexin in the root under diseased and disease-free conditions. Therefore, they can be used expression markers.

• World Health Organization (WHO) has defined a biomarker (Strimbu K, Tavel JA. What are biomarkers? Curr Opin HIV AIDS. 2010 Nov;5(6):463-6. doi: 10.1097/COH.0b013e32833ed177. PMID: 20978388; PMCID: PMC3078627.). What is the purpose with the authors use the term “biomarker” here?

• Define the abbreviation “γARD”

• What do you mean by ARD variants?

• How many plants were used for qRT-PCR? How were these plants grown; by sexual means or by asexual means? Was there any zygosity difference among the plants?

• Whether the plants grown under the two conditions were genotypically comparable or same?

• The details on the number of plant samples, technical replicates and the biological replicates used in the study need be to more clear.

• PCR efficiency and melting curve analysis should be provided.

• Under conclusions, it is stated that “Their expression was strongly and consistently induced in roots of apple plants grown in three different soils affected by ARD.”. This is in comparison to what? Authors should clearly mention what they are comparing each time. Otherwise, it becomes difficult to understand.

• Overall, the text should be revised to make it readable and simple.

6. PLOS authors have the option to publish the peer review history of their article (what does this mean?). If published, this will include your full peer review and any attached files.

Reviewer #1: Yes: Yanmin Zhu, Research Molecular Biologist, USDA ARS Wenatchee WA 98801

Reviewer #2: No

Reviewer #3: Yes: R. S. Bhat

---

## [Author Response · Author response to Decision Letter 0]

16 Jul 2020

Dear valued Editor and Reviewers, 

please find our response attached as "Response to Reviewers". Thank you!

---

## [Decision Letter · Decision Letter 1]

27 Jul 2020

PONE-D-20-08054R1

Identification and validation of early genetic biomarkers for apple replant disease

PLOS ONE

Dear Dr. Rohr,

Thank you for submitting your manuscript to PLOS ONE. After careful consideration, we feel that it has merit but does not fully meet PLOS ONE’s publication criteria as it currently stands. Therefore, we invite you to submit a revised version of the manuscript that addresses the points raised during the review process.

ACADEMIC EDITOR:

The manuscript has been revised except for few more minor suggestions for improvement in presentation.

We look forward to receiving your revised manuscript.

Kind regards,

Suprasanna Penna

Academic Editor

PLOS ONE

Additional Editor Comments (if provided):

The manuscript has been revised except for few more minor suggestions for improvement in overall presentation.

Reviewers' comments:

Reviewer's Responses to Questions

**Comments to the Author**

1. If the authors have adequately addressed your comments raised in a previous round of review and you feel that this manuscript is now acceptable for publication, you may indicate that here to bypass the “Comments to the Author” section, enter your conflict of interest statement in the “Confidential to Editor” section, and submit your "Accept" recommendation.

Reviewer #1: All comments have been addressed

Reviewer #3: All comments have been addressed

2. Is the manuscript technically sound, and do the data support the conclusions?

Reviewer #1: Yes

Reviewer #3: Yes

3. Has the statistical analysis been performed appropriately and rigorously? 

Reviewer #1: Yes

Reviewer #3: Yes

4. Have the authors made all data underlying the findings in their manuscript fully available?

Reviewer #1: Yes

Reviewer #3: Yes

5. Is the manuscript presented in an intelligible fashion and written in standard English?

Reviewer #1: Yes

Reviewer #3: Yes

6. Review Comments to the Author

Reviewer #1: The comments and questions raised by review have been addressed. Therefore, the current form of manuscript is ready to be accepted for publication.

Reviewer #3: • Instead of stating lie “Apple replant disease (ARD) is a serious threat to producers of apple trees and fruits worldwide.”, you can say Apple replant disease (ARD) is a serious disease in apple which causes ---. Indicate the extent of loss, damage.

• Basically, the authors need to highlight how BIS3 and B4Hb express under disease condition in comparison to disease-free condition.

• Also, they need to establish that the expression of BIS3 and B4Hb genes is very specific (express only in the presence of the pathogen).

• Instead of defining the disease in terms of the soils, the authors can define the disease In terms of the presence or absence of the pathogen.

• Is this method of biomarker detection a destructive method?

7. PLOS authors have the option to publish the peer review history of their article (what does this mean?). If published, this will include your full peer review and any attached files.

Reviewer #1: **Yes: **Yanmin Zhu, Research Molecular Biologist, USDA ARS Tree Fruit Research Lab, Wenatchee WA USA

Reviewer #3: **Yes: **Ramesh S. Bhat

---

## [Author Response · Author response to Decision Letter 1]

7 Aug 2020

Dear Editor and Reviewers, 

please find our responses to the points you raised in the attached file "Response to Reviewers". 

Thank you!

---

## [Decision Letter · Decision Letter 2]

26 Aug 2020

Identification and validation of early genetic biomarkers for apple replant disease

PONE-D-20-08054R2

Dear Dr. Rohr,

We’re pleased to inform you that your manuscript has been judged scientifically suitable for publication and will be formally accepted for publication once it meets all outstanding technical requirements.

Kind regards,

Suprasanna Penna

Academic Editor

PLOS ONE

Additional Editor Comments (optional):

Authors have revised the manuscript as per reviewers suggestions and comments.

Reviewers' comments:

Reviewer's Responses to Questions

**Comments to the Author**

1. If the authors have adequately addressed your comments raised in a previous round of review and you feel that this manuscript is now acceptable for publication, you may indicate that here to bypass the “Comments to the Author” section, enter your conflict of interest statement in the “Confidential to Editor” section, and submit your "Accept" recommendation.

Reviewer #3: All comments have been addressed

2. Is the manuscript technically sound, and do the data support the conclusions?

Reviewer #3: Yes

3. Has the statistical analysis been performed appropriately and rigorously? 

Reviewer #3: Yes

4. Have the authors made all data underlying the findings in their manuscript fully available?

Reviewer #3: Yes

5. Is the manuscript presented in an intelligible fashion and written in standard English?

Reviewer #3: Yes

6. Review Comments to the Author

Reviewer #3: Content is fine now. It can be published.

7. PLOS authors have the option to publish the peer review history of their article (what does this mean?). If published, this will include your full peer review and any attached files.

Reviewer #3: **Yes: **Bhat, R. S.

---

## [Editor Report · Acceptance letter]

16 Sep 2020

PONE-D-20-08054R2 

Identification and validation of early genetic biomarkers for apple replant disease 

Dear Dr. Rohr:

I'm pleased to inform you that your manuscript has been deemed suitable for publication in PLOS ONE. Congratulations! Your manuscript is now with our production department. 

Kind regards, 

on behalf of

Dr. Suprasanna Penna 

Academic Editor

PLOS ONE